# Current Status of the Spectrum and Therapeutics of *Helicobacter pylori*-Negative Mucosa-Associated Lymphoid Tissue Lymphoma

**DOI:** 10.3390/cancers14041005

**Published:** 2022-02-16

**Authors:** Sung-Hsin Kuo, Kun-Huei Yeh, Chung-Wu Lin, Jyh-Ming Liou, Ming-Shiang Wu, Li-Tzong Chen, Ann-Lii Cheng

**Affiliations:** 1Department of Oncology, National Taiwan University Hospital, National Taiwan University College of Medicine, Taipei 100, Taiwan; shkuo101@ntu.edu.tw (S.-H.K.); khyeh@ntu.edu.tw (K.-H.Y.); 2Cancer Research Center, National Taiwan University College of Medicine, Taipei 100, Taiwan; 3Graduate Institute of Oncology, National Taiwan University College of Medicine, Taipei 100, Taiwan; 4Department of Pathology, National Taiwan University Hospital, National Taiwan University College of Medicine, Taipei 100, Taiwan; chungwulin@ntu.edu.tw; 5Department of Internal Medicine, National Taiwan University Cancer Center, National Taiwan University College of Medicine, Taipei 106, Taiwan; dtmed046@pchome.com.tw; 6Department of Internal Medicine, National Taiwan University Hospital, National Taiwan University College of Medicine, Taipei 100, Taiwan; mingshiang@ntu.edu.tw; 7National Institute of Cancer Research, National Health Research Institutes, Tainan 704, Taiwan; leochen@nhri.org.tw; 8Department of Internal Medicine, Kaohsiung Medical University Hospital, Kaohsiung Medical University, Kaohsiung 807, Taiwan; 9Department of Internal Medicine, National Cheng-Kung University Hospital, Tainan 704, Taiwan; 10Department of Oncology, National Taiwan University Cancer Center, National Taiwan University College of Medicine, Taipei 106, Taiwan

**Keywords:** MALT, *Helicobacter pylori*, negative, spectrum, bacteria, lymphomagenesis, macrolides, immunomodulatory drugs

## Abstract

**Simple Summary:**

The prevalence of *Helicobacter pylori* (HP)-negative gastric mucosa-associated lymphoid tissue (MALT) lymphoma has increased over the last two decades, whereas that of HP-positive gastric MALT lymphoma has decreased. Although the role of first-line antibiotics in the treatment of HP-negative gastric MALT lymphomas remains ambiguous, several case series have reported that a first-line HP eradication therapy (HPE)-like regimen could result in complete remission in a proportion of patients with localized HP-negative gastric MALT lymphoma. Previous sporadic reports have indicated that certain patients with extragastric MALT lymphoma can respond to first-line antibiotic treatment as well. These findings suggest that, in contrast to antibiotic-unresponsive tumors, antibiotic-responsive tumors may be recognized within the spectrum of HP-negative MALT lymphoma. In addition to conventional chemotherapy and immunochemotherapy, macrolide antibiotics and immunomodulatory drugs have been previously used and demonstrated to be efficacious. This article provides the spectrum and therapeutics for HP-negative MALT lymphoma.

**Abstract:**

*Helicobacter pylori* (HP)-unrelated mucosa-associated lymphoid tissue (MALT) lymphoma includes the majority of extragastric MALT lymphomas and a small proportion of gastric MALT lymphomas. Although the role of first-line antibiotics in treating HP-negative gastric MALT lymphomas remains controversial, HP eradication therapy (HPE)-like regimens may result in approximately 20–30% complete remission (CR) for patients with localized HP-negative gastric MALT lymphoma. In these patients, *H*. *heilmannii*, *H. bizzozeronii*, and *H. suis* were detected in sporadic gastric biopsy specimens. Extragastric MALT lymphoma is conventionally treated with radiotherapy for localized disease and systemic chemotherapy for advanced and metastatic diseases. However, a proportion of extragastric MALT lymphomas, such as ocular adnexal lesions and small intestinal lesions, were reported to be controlled by antibiotics for *Chlamydophila psittaci* and *Campylobacter jejuni*, respectively. Some extragastric MALT lymphomas may even respond to first-line HPE. These findings suggest that some antibiotic-responsive tumors may exist in the family of HP-negative MALT lymphomas. Two mechanisms underlying the antibiotic responsiveness of HP-negative MALT lymphoma have been proposed. First, an HPE-like regimen may eradicate the antigens of unknown bacteria. Second, clarithromycin (the main component of HPE) may have direct or indirect antineoplastic effects, thus contributing to the CR of these tumors. For antibiotic-unresponsive HP-negative MALT lymphoma, high-dose macrolides and immunomodulatory drugs, such as thalidomide and lenalidomide, have reported sporadic success. Further investigation of new treatment regimens is warranted.

## 1. Introduction

Isaacson and Wright first illustrated an entity of low-grade B-cell lymphoma in which most lymphoma cells were located in Peyer’s patch-resembling structures and had a similar histomorphological appearance to mucosa-associated lymphoid tissue (MALT). This type of B-cell lymphoma was found in extranodal organs, such as the stomach, ocular adnexa, salivary glands, lungs, thyroid gland, and intestinal tract [1]. According to the Revised European-American Lymphoma classification system of the World Health Organization, this entity of MALT lymphoma is a distinct subgroup of indolent B-cell lymphomas and is renamed as the extranodal marginal zone lymphoma (MZL) of the MALT type [2,3]. Extranodal MZL or MALT lymphoma is the most common extranodal indolent B-cell lymphoma histologically characterized by diffuse small- and medium-sized lymphocytes (also called centrocyte-like cells), lymphoepithelial lesions, and partial plasma cell differentiation [4,5]. Among these extranodal MALT lymphomas, gastric MALT lymphomas are the most common, and are closely linked to *Helicobacter pylori* (HP) infection [5,6]. Several studies have demonstrated that first-line HP eradication therapy (HPE) can result in approximately 60–80% complete remission (CR) of localized gastric MALT lymphoma [6,7,8]. However, approximately 20% of patients with gastric MALT lymphoma are initially diagnosed with HP-unrelated diseases, and thus have a lower CR rate in response to first-line HPE [9,10]. In addition to HP-negative gastric MALT lymphoma, extragastric MALT lymphomas have been described in various extranodal mucosal sites, including the conjunctiva, orbit, small intestine, large intestine, salivary gland, thyroid, skin, lung, breast, bladder, esophagus, and liver [10,11,12]. Extragastric MALT lymphomas are commonly characterized by HP-unrelated lymphoma [10,11,12]. 

In contrast to HP-positive gastric MALT lymphoma, the use of antibiotics as the first-line treatment for localized HP-negative gastric MALT lymphoma remains controversial [13,14]. However, some investigators have shown that approximately 20–30% of patients with early-stage HP-negative gastric MALT lymphoma respond well to first-line HPE-like regimens [9,10,15,16]. Furthermore, studies have reported that some HP-negative *Helicobacter* spp., such as *H. heilmannii*, *H. suis*, and *H. bizzozeronii*, were detected in sporadic gastric biopsy samples of antibiotic-responsive HP-negative gastric MALT lymphoma [17,18]. In addition, increasing evidence demonstrates that *Campylobacter jejuni* and *Chlamydia psittaci* infections are closely linked to immunoproliferative small intestine disease (IPSID, small intestinal MALT lymphoma) and ocular adnexal MALT lymphoma (OAML), respectively [10,12,19]. Therefore, the administration of antibiotics to eradicate *C. jejuni* and *C. psittaci* cures a proportion of these patients [10,12,19]. 

However, for extragastric MALT lymphomas that are not associated with bacterial infection, the use of radiotherapy (RT) as first-line treatment for localized disease and administration of systemic therapy, including chemotherapy or rituximab for advanced and metastatic diseases, remains the standard therapy [10,20,21]. Considering that the biological behaviors of localized HP-negative gastric MALT lymphoma and extragastric MALT lymphoma are relatively indolent, alternative treatments with less toxic strategies for treating these diseases have been explored [22,23,24,25]. Anecdotal case reports revealed that some patients with HP-negative extragastric MALT lymphomas might respond well to first-line HPE and are cured by antibiotic treatment [22]. These findings suggest that some antibiotic-responsive extragastric MALT lymphomas may be recognized within the spectrum of antibiotic-responsive HP-negative MALT lymphoma. In addition to conventional systemic chemotherapy, high-dose clarithromycin, a macrolide, and an immunomodulatory agent, lenalidomide, have been reported to produce sporadic success for antibiotic-unresponsive or systemic treatment-refractory MALT lymphoma [24,25].

This article describes the treatment results of first-line antibiotics, including HPE regimens, in patients with localized HP-unrelated MALT lymphoma and the possible mechanisms of treatment success. Furthermore, we present the treatment efficacies and underlying antitumor biological machinery of high-dose macrolides, including clarithromycin and azithromycin, and immunomodulatory drugs, such as thalidomide and lenalidomide, in antibiotic-unresponsive, relapsed, and systemic treatment-refractory MALT lymphomas. 

## 2. Patients with HP-Negative Gastric MALT Lymphoma Who Respond Well to First-Line HPE

In a systematic review of published data on the clinical symptoms and endoscopic manifestations of gastric lymphoma, including MALT lymphoma and diffuse large B-cell lymphoma (DLBCL) with and without MALT lymphoma, Zullo et al. reported that the HP infection status was negative in 117 (11.2%) of 1146 cases of gastric lymphoma [26]. However, the widespread administration of the HPE regimen for patients with HP-positive gastritis or ulceration may prevent HP-associated gastric diseases, such as gastric cancer, and may alter the prevalence of HP infection in patients with newly diagnosed gastric MALT lymphoma [27,28,29]. For example, Raderer et al. revealed that the HP-negative infection rates were 18% and 31.2% in patients with gastric MALT lymphoma diagnosed before and after 2014, respectively [30]. Mendes L et al. also showed that among gastric MALT lymphoma patients, the prevalence of HP-negative infection was significantly higher in patients diagnosed between 2005 and 2013 than in those diagnosed before 2005 (25/37 cases [67.6%] vs. 24/61 cases [39.3%]) [31]. Kuo et al. also reported that HP-negative infection was found in 29 (31.5%) of 92 patients with gastric MALT lymphoma diagnosed between 2005 and 2014 [16].

Although earlier studies showed that most patients with HP-negative gastric MALT lymphoma lacked tumor remission after receiving antibiotic treatment [13,32,33], several case series reported that some patients with HP-negative MALT lymphoma were responsive to first-line HPE treatment [9,10,15,16,30,34,35,36] (Table 1). In a pooled analysis of 110 patients obtained from 11 studies in which HP-negative status was confirmed by at least three negative HP examinations, Zullo et al. revealed a CR rate of 15.5% (17/110) for HP-negative gastric MALT lymphoma with HPE [15]. In contrast, Raderer et al. reported that the CR rate of first-line HPE for treating HP-negative gastric MALT lymphoma was 38.5% (5/13) [30]. In their study, most patients achieved CR within 9 months after HPE, except one patient who achieved partial remission (PR) at 23 months and subsequently achieved CR at 36 months after HPE (Table 1) [30]. Furthermore, three series with more than 15 patients with HP-negative gastric MALT lymphoma showed that first-line HPE resulted in CR rates of 29.4% (5/17) [37], 57.1% (16/28, the median time to CR was 11.5 months [range 10.0 to 22.3]) [38], and 23.1% (6/26, the median time to CR was 2 months [range 1–6]) [39], respectively (Table 1). Our findings revealed that among 25 patients with HP-negative gastric MALT lymphoma whose HP was negative for all histologic examination, rapid urease test, C13 breath test, and serology examination, eight patients (32%) achieved CR, and the median time to CR was 7 months (range, 1–24 months) for these responders after first-line HPE (Table 1) [16].

The divergence between the low CR rate with antibiotics for HP-negative gastric MALT lymphoma in earlier studies [13,14,15,32,33,34,35] and the modest CR rate HPE in other results [16,30,36,37,38,39] may be due to the insufficient time allowed lymphoma regression. It is known that some patients may achieve CR after 9 months of completing HPE, and some as late as 24 months [16,30,38]. Another reason is that t(11;18)(q21;q21), a marker for antibiotic unresponsiveness [40,41], may be higher in series reporting a lower CR rate than those with a higher CR rate. We previously reported that t(11;18)(q21;q21)/BIRC3-MALT1 was found in 7 (43.8%) of 16 patients with HP-negative gastric MALT lymphoma who did not respond to antibiotic treatment [16]. In a recent study of genetic alterations and somatic mutations in 57 patients with HP-negative gastric MALT lymphoma, Kiesewetter reported that 22 patients (38.6%) had MALT1 translocation (most t(11;18)(q21;q21)/BIRC3-MALT1), and 14 patients had mutations in NF-κB signaling molecules, such as TNFAIP3, CARD11, and MAP3K14 [42]. These findings may explain why none of the nine patients with HP-negative gastric MALT lymphoma who received first-line antibiotics achieved CR [42] because NF-κB signaling is often associated with antibiotic unresponsiveness in gastric MALT lymphoma [43,44,45].

## 3. Non-*Helicobacter pylori* Helicobacter (NHPH) May Be Associated with Antibiotic-Responsiveness of HP-Negative Gastric MALT Lymphoma

Although the reason for responsiveness to first-line HPE in 20%–30% of patients with HP-negative gastric MALT lymphoma remains uncertain, previous studies suggest that NHPH, such as *H. heilmannii*, is linked to the development of gastric MALT lymphoma, although its prevalence is very low [17,46,47]. In a comparison of 202 patients with HP-associated gastritis and ulcers, and 202 patients with *H. heilmannii*-associated gastritis, Stolte et al. revealed that *H. heilmannii* was rarely associated with intestinal metaplasia and MALT when compared with HP bacteria [46]. Notably, among these 202 patients, seven had simultaneous gastric MALT lymphoma [46]. Furthermore, Morgner et al. reported that five patients with *H. heilmannii*-positive gastric MALT lymphoma confirmed through 16S ribosomal RNA amplification and sequencing methods achieved CR after receiving omeprazole and amoxicillin [17]. In a pathological review of gastric biopsies from 4074 patients, Okiyama et al. identified 11 patients with chronic gastritis, and four patients with gastric MALT lymphoma had *H. heilmannii* [47], which manifested as predominant straight appearances and large sizes in histological morphology, as previously described by Helimann and Borchard [48]. Among these four patients with *H. heilmannii*-positive gastric MALT lymphoma, two patients underwent antibiotic treatments, including lansoprazole, amoxicillin, and clarithromycin, and subsequently achieved CR [47]. These findings indicate that the HPE regimen can eradicate *H. heilmannii*, and thus cure patients with *H. heilmannii*-associated HP-negative gastric MALT lymphoma.

Nakamura et al. analyzed 236 HP-negative cases with gastric disease but without a history of HPE, in which the HP-negative status was confirmed by negative results in rapid urease test, histological examination, and polymerase chain reaction (PCR) for detecting HP [49]. They found that 49 cases were positive for NHPH, including *Helicobacter suis* (*n* = 20), *H. heilmannii sensu stricto**/H. ailurogastricus* (Hhss/Ha) (*n* = 7), and non-Hhss/Ha (*n* = 22) [49]. Among 49 cases of NHPH, 11 were diagnosed with gastric MALT lymphoma (four with *H. suis*, two with Hhss/Ha, and five with non-Hhss/Ha) [49]. In a recent analysis of first-line HPE in 182 patients with gastric MALT lymphoma, Takigawa et al. showed that HP-negative patients had a higher prevalence rate of NHPH than HP-positive patients through *Helicobacter* sp.-specific PCR assay (16/29 [55%] vs. 3/29 [10%], *p* < 0.05) [18]. Among 16 HP-negative NHPH-positive cases, five were positive for *H. suis*, eight for *H. bizzozeronii*, and three for both *H. suis* and *H. bizzozeronii* [18]. Furthermore, among patients with neither HP infection nor BIRC3-MALT1 fusion protein, positive NHPH cases had a higher CR rate than those with negative NHPH (12/16 [75%] vs. 3/13 [23%], *p* = 0.0092) [18]. These findings suggest that NHPHs are linked with the development of certain HP-negative gastric MALT lymphomas, and eradication of NHPH by first-line HPE can eradicate this subtype of gastric MALT lymphoma. Further large studies to assess the prevalence of NHPH in patients diagnosed with HP-negative gastric MALT lymphoma are warranted. 

Although most NHPHs are negative for the rapid urease test and ^13^C urea breath test, several studies have shown that certain NHPHs were positive for urease test and contaminated with HP in HP-positive patients [50,51,52]. Goji et al. reported that a patient presenting with nodular gastritis at the antrum tested positive for rapid urease test and a ^13^C urea breath test, but had negative results for stool antigen and serum anti-HP IgG antibody, and was further diagnosed with *H. suis* infection through PCR analysis of the 16S rRNA [50]. Goji et al. also reviewed the sensitivities of methods for detecting *H. heilmannii*-like organisms from 26 articles, and demonstrated that rapid urease test and immunohistochemical analysis were 40%, whereas urea breath test, blood antibody analysis, and stool antigen analysis were 14.8%, 23.1%, and 0%, respectively [50]. Yakoob et al. assessed the prevalence of NHPH and HP in 250 patients with dyspepsia through PCR analyses of *Helicobacter* genus-specific 16S rDNA, and reported that coinfection of *H. heilmannii* with HP was 6% in 17 patients, and the coinfection of *H. felis* with HP was 4% in 10 patients [51]. In the assessment of the prevalence of *H. heilmannii* sensu lato *(H. heilmannii* s.l.), a group of NHPH species [53], including *H. suis*, *H. felis*, *H. bizzozeronii*, *H. heilmannii* sensu stricto (s.s.), and *H. salomonis* in samples of rapid urease test positive gastric biopsy using *H. heilmannii* s.l.-specific PCR followed by nucleotide sequencing, Liu et al. showed that NHPH infection was found in 178 (11.9%) of 1499 HP-positive patients [52]. Takigawa et al. also showed that 10% of patients with HP-positive MALT lymphoma had co-infection with NHPH, and NHPH-positive patients had more endoscopic nodular appearances in the gastric mucosa than NHPH-negative cases (3/16 [18.9%] vs. 0/29 [0%]) did [18]. These studies suggest that certain portions of NPHP may be involved in lymphomagenesis of urease test-positive patients, especially for patients with MALT lymphoma who presented with nodular appearances and negative serum antibody and stool antigen examination [18,50,54]. Although previous clinical studies exploring the efficacy of HPE in treating HP-positive gastric MALT lymphoma did not examine the prevalence of NHPH in these patients [6,7], further studies investigating the occurrence of NHPH in patients with MALT lymphoma presenting with nodular-like appearance irrespective of HP status are warranted.

## 4. Efficacies of First-Line Antibiotics Treatment for Extragastric MALT Lymphoma

Unlike localized gastric MALT lymphoma, which is mainly treated with first-line HPE, the optimal management for localized extragastric MALT lymphoma has yet to be elucidated [10,12]. Conventionally, most patients with localized extragastric MALT lymphoma are treated with RT [20,55]. Considering the indolent biological behavior of localized extragastric MALT lymphoma, the use of less toxic strategies, such as “antibiotics,” as a frontline treatment for this subtype of MALT lymphoma has been explored [19,20,22].

In addition to gastric MALT lymphoma, which is significantly associated with HP infection, increasing evidence demonstrates that *C. jejuni*, *C. psittaci*, and *Borrelia burgdorferi* are associated with IPSID, OAML, and cutaneous MALT lymphoma, respectively [56,57,58,59,60]. IPSID, an endemic disease, mainly occurs in the Mediterranean and sporadically in Western and Asian countries [10,20,56,61]. Patients with IPSID often present with chronic diarrhea complicated by malabsorption and weight loss, and are treated with first-line antibiotics, including tetracycline, metronidazole, ampicillin, and corticosteroids [61]. 

Ferreri et al. first reported an association between *C. psittaci* DNA in tumor samples and patients with OAML [57]. They demonstrated that using doxycycline to eradicate *C. psittaci* can cure a proportion of patients with OAML [57]. In a pooled analysis for the prevalence of *C. psittaci* DNA in patients with ocular lymphoma, Travaglino et al. revealed that *C. psittaci* was diverse in different countries, ranging from a low rate of 0% to 5% (Japan and Unities status) to a high rate of 33% to 50% (Austria, Italy, and Korea) [62]. In their analyses, the most common subtype of ocular adnexal B-cell lymphomas associated with *C. psittaci* infection was MALT lymphoma subtypes, with an odds ratio of 2.183 (95% confidence interval, 1.092–4.360) [62]. In a systematic analysis of the efficacy of doxycycline in eradicating *C. psittaci* in 131 patients with OAML (from four retrospective studies and three prospective series), the authors demonstrated that first-line antibiotics resulted in a CR rate of 17.6% (*n* = 23) and a partial remission (PR) rate of 27.5% (*n* = 36) [19,22,57,63]. In an endemic area of Korea, Han et al. reported that among 90 patients with OAML who received one or two courses of first-line doxycycline (100 mg bid for 3 weeks each course), four patients achieved CR, 20 achieved PR, and 34 had stable disease [64]. Han et al. further found that the T1 stage was significantly associated with the overall response rate (ORR) when compared with T2 to T4 stages (37% [21/57] vs. 10% [3/28]), *p* < 0.001) [64]. Although most ORRs were observed in *C. psittaci*-positive patients, some OAML patients without *C. psittaci* infection had lymphoma regression after doxycycline treatments [19,22,57,63]. These findings suggest that doxycycline may eradicate non-*C. psittaci* microorganisms, and these doxycycline-sensitive unknown bacteria may be associated with the lymphomagenesis of OAML.

Patients with cutaneous MALT lymphoma often present with asymptomatic single papules, plaques, or nodules before diagnosis, and certain patients demonstrate serum antibody positivity for *B. burgdorferi* [60]. In endemic areas of Europe, the prevalence of *B. burgdorferi* was approximately 10% to 42% in patients with cutaneous MALT lymphoma through PCR analyses of *B. burgdorferi* DNA [59]. Based on the indolent course of most patients with cutaneous MALT lymphoma, the administration of first-line antibiotics using cephalosporins and tetracyclines is reasonable based on case reports demonstrating the antibiotic responsiveness of *B. burgdorferi*-positive cutaneous MALT lymphoma [59,60,65]. In a meta-analysis of 506 patients with primary cutaneous lymphoma obtained from 10 studies, Travaglino et al. showed that the prevalence of *B. burgdorferi* infection confirmed by PCR assay of *B. burgdorferi* DNA was significantly associated with B-cell lymphoma, including MALT lymphoma, and the positivity rate for *B. burgdorferi* in primary cutaneous MALT lymphoma was 8.3% [66]. 

Patients with pulmonary MALT lymphoma frequently present with an asymptomatic and indolent disease course, and most patients are diagnosed during an annual checkup [20,22,67]. Unlike the close association between bacterial infections and IPSID, OAML, and cutaneous MALT lymphoma, the association between *Achromobacter*
*xylosoxidans* and pulmonary MALT lymphoma is controversial [68,69,70]. Adam et al. first reported that the prevalence of *A. xylosoxidans* was higher in tumor samples from patients with pulmonary MALT lymphoma than in lung tissues of control cases from six European countries (46.0% (57/124) vs. 18.3% (15/82), *p* = 0.004) through a specific nested PCR approach for the DNA of *A. xylosoxidans* [68]. Another study conducted by Borie et al. in France showed that sequences of *A. xylosoxidans* DNA were detected in four (30.8%) of thirteen patients with MALT lymphoma and four (40.0%) of ten healthy controls [69]. However, in the Japanese population, Aoyama et al. revealed that *A. xylosoxidans* DNA was only detected in one (1.9%) of fifty-two patients with pulmonary MALT lymphomas [70]. Considering that pulmonary MALT lymphoma is relatively indolent and *A. xylosoxidans* is relatively resistant to antibiotic treatment, close observation of localized pulmonary MALT lymphoma has been suggested [71]. 

Unlike IPSID, OAML, cutaneous, and pulmonary MALT lymphomas that are reported to be associated with bacterial infection, few reports have shown an association between infectious bacteria and other extragastric MALT lymphomas. In a study of the relationship between *C. psittaci* DNA in blood samples and patients with salivary gland MALT lymphoma and Sjogren’s syndrome, Fabris et al. showed that the detection rate for *C. psittaci* was higher in patients with MALT lymphoma (6/18, 33.3%) than in those without (5/56, 8.9%, *p* = 0.012) [72]. However, no reports have demonstrated the efficacy of doxycycline in the treatment of salivary gland MALT lymphoma. 

However, anecdotal case series from the reviews of first-line antibiotics for extragastric MALT lymphoma by Kiesewetter et al. showed that two cases of salivary gland, one case of thyroid gland, and two cases of bladder MALT lymphoma achieved CR after receiving HPE and remained lymphoma-free after a median follow-up of 12 months (ranging from 5 to 48 months), in which gastric biopsy showed the presence of HP infection [22]. Won et al. reviewed the clinicopathological features and CR rates of 67 patients with colorectal MALT lymphoma who received differential first-line treatment, including surgery, chemotherapy, radiotherapy, and antibiotic treatment (including HPE), Won et al. revealed that 12 (80.0%) of 15 patients achieved CR after receiving antibiotic treatment, and 11 (91.7%) of 12 patients had no recurrence after a long-term follow-up [73]. Moriya et al. also reported that a patient with esophageal MALT lymphoma achieved CR after HPE (positive anti-HP immunoglobulin G antibody in serum) and remained lymphoma-free, although HP infection in esophageal histological specimens was absent [74]. 

In addition to HP-negative gastric MALT lymphoma, some patients with extragastric MALT lymphoma may even respond to first-line HPE. These findings indicate that some antibiotic-responsive MALT lymphomas may exist in the family of HP-negative MALT lymphomas. However, the molecular mechanisms underlying these antibiotic-responsive HP-negative MALT lymphomas remain unclear.

## 5. Possible Molecular Mechanisms of Antibiotic-Responsive HP-Negative MALT Lymphoma

In addition to NHPH, other unknown bacterial sources can promote lymphomagenesis in HP-negative MALT lymphomas. Several hypotheses explain why certain patients with HP-negative MALT lymphoma respond to first-line HPE: (1) HPE regimens may eradicate NHPH, which is associated with the growth of HP-negative gastric MALT lymphoma in humans; (2) an HPE-like regimen may eradicate bacteria-associated antigens of unknown bacteria or unknown bacteria that may be associated with the lymphomagenesis of HP-negative MALT lymphoma; (3) clarithromycin, the principle component of the HPE regimen, may have direct or indirect antineoplastic effects, and thus cause CR of these tumors (Figure 1).

Several lines of evidence have demonstrated that the indirect lymphomageneses of HP-positive gastric MALT lymphoma are mainly contributed by the communication with B-cells by tumor-infiltrating T cells and their stimulating T-helper (Th) cytokines, chemokines, costimulatory molecules, and regulatory T cells (Foxp3), or by micropathogenic antigenic stimulation or other unknown antigen stimuli [75,76,77]. These factors may explain why HPE may diminish the efficacy of B-cell growth from T cells and antigen stimuli by eliminating unknown bacteria. Another possible reason is that recruitment of HP-specific T-cells and antigens to tumor sites and their microenvironments from sites of HP-infected stomachs can be diminished through HPE [78], because some patients with extragastric MALT lymphoma who achieved CR after HPE had HP infection in the stomach, although HP could not be detected in tumor sites [22,74,79] (Figure 1).

For these antibiotic-responsive HP-negative MALT lymphomas, we cannot exclude the possibility that unknown bacteria, which HPE may eradicate, could participate in the formation of this tumor subtype. In the metagenomic analyses of microbiota obtained from the mucosal biopsy specimens of 33 patients with gastric MALT lymphoma (24 with naïve HP-negative status and 9 with previous HPE history) and 27 control patients without HP infection and cancer, Tanaka et al. demonstrated that the genera *Burkholderia* and *Sphingomonas* were more frequently found in patients with MALT lymphoma than in those of the control group [80]. These findings indicate that these microbes may be involved in the development of HP-negative MALT lymphomas. *Burkholderia*, a plant pathogen, has been reported to be an unscrupulous microbiota in immunocompromised patients [81]. Moreover, lectins from *Burkholderia*
*cenocepacia* were found to interact and stimulate B cells to express N-linked glycan-containing B-cell antigen receptors in follicular cell lymphoma [82]. In assessing bismuth-containing HPE regimen in the treatment efficacies of HP and other gastric microbiota, Niu et al. revealed that the genera *Rhodococcus*, *Lactobacillus*, and *Sphingomonas* were more responsive to HPE [83]. These findings indicate that *Sphingomonas* may be linked with HP-negative gastric MALT lymphoma by modulating natural killer (NK) cell function and altering systemic immunity in in vivo mouse studies (Figure 1) [84]. Our previous study also demonstrated that increased infiltration of NK cells was associated with the lymphomagenesis of antibiotic-responsive gastric high-grade transformed MALT lymphoma (renamed as DLBCL with MALT) [85].

Clarithromycin, a macrolide antibiotic, is commonly used as the main component of the HPE regimen [6,7,8]. In addition to eradicating bacteria, clarithromycin has anti-neoplastic effects in B-cell lymphoma cells derived from BALB/c mice by downregulating anti-apoptotic molecules, BCL-2, and upregulating TNFR1 expression and its related molecules, such as caspases-3, -8, and -9 [86]. Clarithromycin was also reported to enhance apoptotic effects on B-lymphocytes isolated from human peripheral blood by activating the FAS/FAS ligand pathway [87]. Additionally, it can enhance apoptotic effects on activated B-lymphocytes by downregulating Bcl-xL expression [88]. Furthermore, clarithromycin attenuated the activation of NF-κB stimulated by TNF-α in U-937 cells (human monocytes), Jurkat cells (T-cell lines), and peripheral blood mononuclear cells [89].

In addition to the direct anti-neoplastic effects of clarithromycin, several studies have demonstrated its immunomodulatory effects, including the activation of NK cells and CD8+ cells and costimulatory molecules, CD80, and alteration of cytokine production [90,91]. In vitro studies of dendritic cells derived from bone marrow progenitor cells showed that clarithromycin upregulated CD80 expression, inhibited IL-6 production, and attenuated IL-2 production in T cells co-cultured with dendritic cells [92]. Considering that IL-6 plays a crucial role in regulating and activating myeloid-derived suppressor cells (MDSCs) and that MDSCs can harbor anti-tumor immunity [93], it is speculated that clarithromycin may have immunomodulatory effects by decreasing IL-6 and subsequently suppressing MDSCs. Furthermore, in in vitro peripheral blood mononuclear cell studies, Ratzinger et al. revealed that azithromycin, another macrolide, and clarithromycin could repress the activity of CD4+ cells by hampering mTOR signaling [94], indicating that macrolides can act as immunomodulatory agents against cancer and lymphoma.

Further exploration of the precise mechanisms of lymphomagenesis for this spectrum of antibiotic-responsive HP-negative MALT lymphoma is warranted because a 2-week antibiotic regimen has fewer adverse effects than conventional radiotherapy or chemotherapy.

## 6. Clinical Efficacy of Radiotherapy for HP-Negative MALT Lymphoma

For localized and antibiotic-refractory HP-negative gastric MALT lymphoma, radiotherapy (RT) can provide local control but may induce acute and chronic adverse effects [10,20,95,96]. For example, patients with OAML who received RT had a high CR rate and promising event-free survival (EFS) but experienced side effects, such as cataracts, dry eyes, and keratitis [95,97]. However, several studies have demonstrated that reduction in RT (field or dose) not only provides promising therapeutic efficacies, but also has lower adverse effects for localized MALT lymphoma [96,98,99,100,101]. In a long-term follow-up of 290 patients with stage IE-IIE HP-independent (antibiotic-refractory or HP-negative) gastric MALT lymphoma who received different RT fields from three consecutive prospective trials of the German Study Group on Gastrointestinal Lymphoma (DSGL), Reinartz et al. revealed that reduced RT field (stage I, involved field [IF] RT with 40 Gy; stage IIE locoregional extended field [EF] RT with 30 Gy followed by boost RT) provided similar EFS and overall survival (OS) when compared with EF RT (stage I, EF or reduced EF; Stage II, EF-mediastinum or EF; 30 Gy followed by 10 Gy boost) [98]. Furthermore, reduced RT field caused less acute hematological toxicities, such as anemia, leukocytopenia, and thrombocytopenia, and gastrointestinal toxicities, such as nausea and diarrhea, and lower frequencies of late toxicities than EF RT [98]. In a retrospective analysis of 178 patients with HP-independent gastric MALT lymphoma, who were stage I (86%) and HP-negative status (80%), and of whom most received involved-site RT (ISRT) with 30 Gy in 15 fractions, Yahalom et al. showed that among 160 patients who received regular panendoscopic examinations, 152 (95%) patients achieved CR, and in most patients acute toxicities were rare, except for two patients who developed grade 3 toxicities [99]. In addition to the reduced RT field, Pinnix et al. revealed that ISRT with 24 Gy (*n* = 11) provided similar local control, freedom from treatment failure, and OS when compared with those receiving ISRT with 30 Gy (*n* = 21) for treating HP-independent gastric MALT lymphoma [100]. In a prospective trial comparing two different IF RTs for stage IE-II1E HP-independent gastric MALT lymphoma, Schmelz et al. showed that RT with 25.2 Gy (*n* = 10) produced a similar CR rate and gastrointestinal adverse effects when compared with RT with 36 Gy (*n* = 12) [101].

In the post-hoc analyses of a randomized, phase 3, non-inferiority trial comparing 24 Gy with 4 Gy for indolent lymphoma, including follicular lymphoma and MZL, authors reported that among 84 patients with MZL, patients (*n* = 41) receiving 24 Gy had a better 5-year local progression-free survival (PFS) than those receiving 4 Gy (*n* = 43) (100% vs. 88%, *p* = 0.015) [96]. Taken together, these findings indicate that reduced field RT with at least 24 Gy could provide promising locoregional control and long-term survival and few adverse effects. However, Pinnix et al. reported that 2 Gy in 2 fractions (total 4 Gy) resulted in a CR of 86% and grade 1 eye syndrome in only 5% of 22 patients with ocular adnexal lymphoma (OAL, including fourteen with MALT lymphoma, five with follicular lymphoma, and two with mantle cell lymphoma) [102]. Cerrato et al. retrospectively analyzed 45 patients with MALT or MZL who were treated with 4 Gy in two 2-Gy fractions as curative or palliative intent, and revealed that low-dose RT (4 Gy) resulted in an ORR of 93% (CR: 51%, PR: 42%) and a 2-year PFS and OS of 76% and 91%, respectively, without causing significant acute and late adverse effects [103]. Baron et al. also showed that among 36 patients with indolent OAL (20 patients with MALT lymphoma), low-dose RT (4 Gy) resulted in a similar ORR rate (100% [CR: 50%] vs. 87.5% [CR 58.3%]) but fewer acute toxicities (6/14 [42.9%] courses vs. 20/24 courses [83.3%], *p* = 0.014) when compared with moderate-dose RT (21–36 Gy) [104]. Prospective studies to validate the local control rate and adverse effects of low-dose RT (4 Gy) in larger patients with MALT lymphoma are warranted.

## 7. Clinical Efficacy of Immunomodulatory Agents for HP-Negative MALT Lymphoma

Systemic therapy is often prescribed in localized patients who cannot tolerate the adverse effects of radiotherapy, or in those with advanced or disseminated MALT lymphoma [10,20,21,23,105]. Previously, several phase II studies evaluating new purine analogs of chemotherapy agents, such as fludarabine and 2CdA, provided approximately 50% ORRs, but these drugs caused myelosuppression [10,20,21,23,105,106]. In a randomized International Extranodal Lymphoma Study Group (IELSG)-19 trial comparing chlorambucil, rituximab, and combined chlorambucil and rituximab as first-line treatments for patients with MALT lymphoma, Zucca et al. showed that the ORR for chlorambucil, rituximab, and combined chlorambucil and rituximab arms were 85.5%, 78.3%, and 94.7%, respectively [107]. The median PFS for chlorambucil, rituximab, and combined chlorambucil and rituximab arm was 8.3 years, 6.9 years, and not reached (*p* = 0.0119), respectively [107]. The 5-year OS was similar in the three arms (chlorambucil, 89%; rituximab, 92%; combined, 90%) [107]. In another phase II study evaluating the treatment response to rituximab and bendamustine (RB), Salar et al. revealed that RB resulted in an ORR of 100% (CR rate of 98%) and a 7-year EFS of 87.7% in 57 patients with HP-independent or relapsed gastric MALT lymphoma, or relapsed or refractory cutaneous MALT lymphoma [108]. Clinical studies have shown that ibrutinib (a Bruton tyrosine kinase inhibitor), a B-cell receptor signaling targeting agent, provides encouraging clinical outcomes in various indolent B-cell lymphomas, including MALT lymphoma [109,110]. In a phase II study exploring the single-agent ibrutinib in patients with relapsed/refractory marginal zone B-cell lymphoma (including nodal, extranodal, and splenic lesions), Noy et al. revealed that ibrutinib provided an ORR of 48% and a median PFS of 14.2 months in 60 patients, whereas 17% of patients discontinued ibrutinib because of its adverse effects [111].

Although rituximab-based chemotherapy provides a high CR rate and promising long-term survival for HP-negative MALT lymphoma [106,107,108,109], several studies have explored the efficacy of chemotherapy-free immunomodulatory agents, such as macrolides, thalidomide, and lenalidomide for these patients, as many have an indolent clinical course and rarely have comorbidities [25,105,112]. In an exploratory study evaluating the safety and efficacy of conventional clarithromycin (500 mg orally, twice daily for 6 months) among 13 patients with relapsed or refractory MALT lymphoma, Govi et al. revealed that clarithromycin resulted in an ORR of 38% (CR, *n* = 2; PR, *n* = 3) and a 3-year PFS of 58% (Table 2) [113]. Based on these encouraging results, the same colleagues designed a phase II trial to assess the efficacy of high-dose clarithromycin (HD-K, clarithromycin 2 g daily, days 1–14, every 21 days, for four courses) [24]. They found that among 23 patients with relapsed or refractory MALT lymphoma, HD-K caused an ORR of 52% (CR, *n* = 6; PR, *n* = 6) and a 2-year PFS of 56% (Table 2) [24]. Ferreri et al. retrospectively analyzed patients with MALT lymphoma who received three different clarithromycin monotherapy regimens (1 g/d, for 6 months; 1 g/d day 1–21, q35 day, for three courses; 2 g/d, days 1–14, q21 days, for four courses), and reported that the ORR and 3-year PFS for those with 1 g/d were 57% and 60%, respectively, and for those with 2 g/d were 41% and 42%, respectively [114]. Regarding the toxicities of clarithromycin, patients receiving 2 g of clarithromycin had more side effects, such as nausea, than those receiving 1 g of clarithromycin [114]. These findings indicate that a daily dose of 1 g of clarithromycin provides an optimal ORR and favorable clinical outcomes for patients with relapsed/refractory MALT lymphoma. 

Azithromycin, a macrolide with a longer half-life than clarithromycin (68 h vs. 5 h), was found to have a greater inhibitory effect on mTOR and CD4+ T cells than clarithromycin in an in vitro study [94,115]. Lagler et al. designed a phase II study to assess the efficacy of azithromycin (1.5 g once every week for 4 weeks) in patients with MALT lymphoma [116]. However, among 16 patients receiving azithromycin (2 with gastric and 14 with extragastric MALT lymphoma), two patients achieved CR, and two had PR (25% ORR) [116]. Furthermore, Scheibenpflug et al. reported that among the 16 patients with MALT lymphoma, high intracellular concentrations of azithromycin in peripheral blood mononuclear cells were not associated with the responsiveness to azithromycin, but intracellular concentrations of azithromycin in polymorphonuclear leukocytes correlated with poor responsiveness [117].

The t(11;18)(q21;q21) translocation, a trigger for NF-κB activation, was more frequently observed in cases of HP-negative MALT lymphoma than in those of HP-positive MALT lymphoma, and NF-κB signaling contributes to the antibiotic unresponsiveness of gastric MALT lymphoma [13,42,43,118,119]. Our previous study also demonstrated that nuclear expression of NF-κB was significantly associated with the antibiotic-unresponsiveness of gastric MALT lymphoma [45,120,121]. Two important immunomodulatory drugs, or “IMiDs,” thalidomide and lenalidomide, have been evaluated for efficacy in certain subgroups of hematological malignancies, including MALT lymphoma, since they have a targeting effect on NF-κB signaling [25,112,122,123]. Although a previous phase II study showed no ORR for thalidomide in the initial eight patients with antibiotic-unresponsive or disseminated MALT lymphoma (Table 2) [124], our previous study revealed that among ten patients with antibiotic-unresponsive or chemotherapy-refractory gastric MALT lymphoma, thalidomide provided an ORR of 50% (CR, *n* = 2; PR, *n* = 3), and a 3-year EFS of 38% (Table 2) [125].

**Table 2 cancers-14-01005-t002:** Summaries of the therapeutic efficacies of immunomodulatory drugs in relapsed or refractory MALT lymphoma.

Authors	Drugs	Number	Age Median (Range)	Lesions	Previous Treatment	ORR	Most Common Side Effects	PFS	Ref
Govi et al.	A regimen *: Clarithromycin 500 mg twice/D for 6 m	13 M/F = 7/6	57 (36–80)	Conjunctival/ ocular+: 11 Gastric: 1 Breast: 1	Relapsed or refractory	38.5% CR: 2 PR: 3	Nausea	3-year 58%	[113]
Ferreri et al.	B regimen *: Clarithromycin 2g/D, D1–14 every 21 D, 4 courses	23 M/F = 5/18	70 (47–88)	Gastric: 3 Extra-gastric or multiple: 20	Relapsed or refractory	52.0% CR: 6 PR: 6	Nausea	2-year 56%	[24]
Ferreri et al.	Clarithromycin 1 g (A regimen * or 500 mg twice daily D 1–21, every 35 D for 3 courses 2 g (B regimen *)	55 M/F = 31/24	65 (30–88)	Gastric: 9 Extra-gastric: 46 Stage I: 40 Stage IV: 15	Tx: naïve *N* = 8 Previous Tx: *N* = 47	47.3% CR: 13 PR: 13 1 g vs. 2 g 57% vs. 41% *p* = 0.28	Nausea 1 g vs. 2 g 25% vs. 52% *p* = 0.03	3-year 52% 1 g vs. 2 g 78% vs. 41% *p* = 0.04	[114]
Lagler et al.	Azithromycin 1500 mg once Weekly 4 time/per m 3 to 6 m	16 M/F = 6/10	68 (47–88)	2 Gastric 14 Extra-gastric	Tx: naïve *N* = 12 Previous Tx: *N* = 4	25% CR: 2 PR: 2	Nausea Diarrhea	No Report	[116]
Troch et al.	Thalidomide 100 mg D initially 200 mg after 4 weeks	8 M/F = 6/2	60 (36–73)	Gastric: 5 Extra-gastric: 3	HPE, fail; or stage IV disease	Initial: 0% 25.0% ** (after Tx) CR, 16.5 m CR, 22.5 m	Neuropathy	No Report	[124,126]
Kuo et al.	Thalidomide 100 mg-200 mg D, for 6 m	10 M/F = 6/2	62 (48–78)	Gastric: 10 I/IIE1: 3 IV: 7	HPE or C/T: failed	50.0% CR: 2 PR: 3	Dizziness	3-year EFS; 38%	[125]
Kiese- wetter et al.	Lenalidomide 25 mg/D D1–21, every 28 D Maximal: 6 cycles	M/F = 8/10	60 (41–79)	Gastric: 5 Extra-gastric: 13	Previous tx: 7 None: 11	61.1% CR: 6 PR: 5	Pruritis	Median FU: 20.3 months 17 pts: alive	[127]
Kiese- wetter et al.	Lenalidomide (Len) 25 mg D, D1–21, every 28 D Maximal: 6 cycles Rituximab (Rit) + Len Len: 20 mg D D1–21 Rit: 375 mg/m^2^. D1 every 28 D Maximal: 8 cycles	Len: 16 R + Len 34 M/F = 19/31	67 (33–85)	Gastric: 16 Extra-gastric: 34 I–II: 33 III–IV: 17	Previous Tx : 24	72.0% CR: 24 PR: 12	Pruritis	54% free-off relapse Median PFS: 72.3 months 5-year OS: 92%	[128]

Abbreviation: ORR, overall response rate; PFS, progression-free survival; Ref, reference; D, daily; m, months; M, male; F, female; Tx, treatment; N, number; CR, complete remission; PR, partial remission; HPE, Helicobacter pylori eradication; EFS, event-free survival; FU, follow-up; OS, overall survival. * Clarithromycin regimen; ** 2 (25%) of 8 patients with thalidomide achieved CR at 16.5 m, and 22.5 m after starting treatment.

In a clinical phase II study assessing the efficacy of single-agent lenalidomide (25 mg daily days 1–21, every 28 days, each cycle) in treating 18 patients with MALT lymphoma (including five with HP-negative gastric and 13 with extragastric lesions), Kiesewetter et al. showed that lenalidomide resulted in an ORR of 61.1% (six with CR and five with PR) [127]. In their study, 17 patients were alive without signs of lymphoma after a median follow-up of 20.3 months (Table 2) [127]. Kiesewetter et al. retrospectively analyzed 25 patients with MALT lymphoma (eighteen with lenalidomide and seven with thalidomide) and showed that seven (28%) patients had delayed-onset responses with a median time to the best response of 7.3 months (range, 5.6–22.5 months) [126]. In an analysis of 50 patients with MALT lymphoma treated with lenalidomide-based regimens (lenalidomide monotherapy, *n* = 16; lenalidomide plus rituximab, *n* = 34), Kiesewetter et al. showed that lenalidomide-based regimens provided an ORR of 74% (54% CR), and estimated the median PFS and the 5-year OS rate to be 72 months and 92%, respectively (Table 2) [128]. Interestingly, among these responders, three patients achieved CR between 12 and 32 months after starting the lenalidomide-based regimen, and one achieved PR from stable disease after 11 months [128]. The most important adverse effects of lenalidomide are mainly non-hematologic, including pruritus, nausea, fatigue, and headache [126,127,128]. Thus, for HP-negative MALT lymphoma, lenalidomide-based regimens can provide promising treatment efficacy and fewer hematological side effects. Physicians should allot sufficient time to observe delayed responses among patients without progression during follow-up after initiating lenalidomide treatment.

Based on the optimal response of single agents, clarithromycin or lenalidomide, the ongoing phase II study designed by the IELSG aims to explore the treatment efficacy and toxicity of combined clarithromycin and lenalidomide (lenalidomide [Revlimid] 20 mg daily day 1–21 and clarithromycin 500 mg twice daily day 1–28, every 28 days for each course) for relapsed or refractory MALT lymphoma (ClinicalTrials.gov Identifier: NCT03031483). Previous studies have reported that low-dose cyclophosphamide has immunomodulatory effects and overcomes the resistance of lenalidomide in patients with multiple myeloma (MM) [129,130,131,132]. Considering the biological relationships between MM and MALT lymphoma (for example, plasma cell differentiation and serum immunoglobulin production) and immunological signaling pathways that contribute to the lymphomagenesis of MALT lymphoma [3,77,133,134,135,136], our ongoing prospective phase II trial (ClinicalTrials.gov Identifier: NCT04604028) aims to assess the ORR, toxicities, and time to progression of combined lenalidomide and low-dose cyclophosphamide (lenalidomide [Leavdo^®^] 15 mg daily, day 1–21; cyclophosphamide 50 mg daily, day 1–21, each cycle every 28 days) for antibiotic-unresponsive and relapsed or refractory MALT lymphoma. Hopefully, these two ongoing trials will elucidate whether relapsed/refractory MALT lymphomas, including HP-negative MALT lymphoma, are responsive to lenalidomide-based combined immunomodulatory agent treatment.

## 8. Conclusions

In addition to HP-gastric MALT lymphoma, specific HP-negative gastric MALT lymphomas respond to first-line HPE. Considering that most patients with gastric MALT lymphoma present with localized disease and an indolent clinical course, the administration of HPE as first-line treatment for newly diagnosed gastric MALT lymphoma without life-threatening emergencies is recommended, because HPE not only eradicates HP and NHPH, but also diminishes antigen stimuli or triggers an immune response against lymphoma cells. Although conventional therapy remains the standard treatment for HP-negative extragastric MALT lymphoma, subtypes of extragastric MALT lymphoma are responsive to specific antibiotics against *C. jejuni*, *C. psittaci*, and *B. burgdorferi*. Interestingly, sporadic reports indicate that certain patients with extragastric MALT lymphoma respond to first-line HPE. In contrast to antibiotic-unresponsive tumors, antibiotic-responsive tumors may be recognized within the spectrum of HP-negative MALT lymphomas. Further identification of unknown bacteria that participate in the lymphomagenesis, and exploration of the precise mechanisms and immune reactions of this spectrum of antibiotic-responsive HP-negative MALT lymphoma, are warranted. Based on the promising results and low toxicities of clarithromycin and lenalidomide for relapsed or refractory MALT lymphoma, further investigations of new chemotherapy-free treatment regimens with high efficacy and minimal adverse effects for antibiotic-unresponsive HP-negative MALT lymphoma are needed.

## Figures and Tables

**Figure 1 cancers-14-01005-f001:**
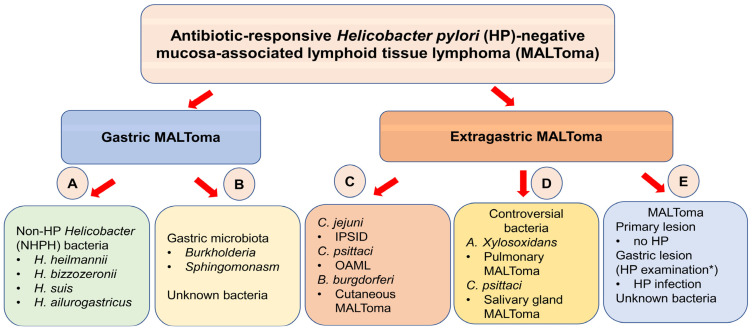
Schema of the spectrum of antibiotic-responsive *Helicobacter pylori* (HP)-negative mucosa-associated lymphoid tissue (MALT) lymphoma. (**A**) Among antibiotic-responsive HP-negative gastric MALT lymphoma, non-HP *Helicobacter* (NHPH) bacteria, such as *H*. *heilmannii**, H. bizzozeronii, H. suis*, and *H. ailurogastricus* can be detected in sporadic cases. However, the precise mechanisms of NHPH on antibiotic-responsiveness remain unclear. (**B**) Gut microbiota, such as genera *Burkholderia*, *Sphingomonasm*, and unknown microbiota, may be involved in the lymphomagenesis of antibiotic-responsive HP-negative gastric MALT lymphoma. (**C**) *C. jejuni*, *C. psittaci*, and *B. burgdorferi* are reported to be associated with the development of immune proliferative small intestinal diseases (IPSID), ocular adnexal MALT lymphoma (OAML), and cutaneous MALT lymphoma, and the use of doxycycline, tetracycline, metronidazole, doxycycline, or cephalosporin to eradicate the aforementioned bacteria, respectively, was found to be efficacious. (**D**) However, the association between *A. xylosoxidans* and the occurrence of pulmonary MALT lymphoma is controversial, as is the association between *C. psittaci* and the development of salivary gland MALT lymphoma. (**E**) Among sporadic cases that respond well to first-line HP eradication therapy (HPE), the presence of HP in the stomach but no direct evidence of HP in the primary lesion may explain the possibility that antigen stimuli or indirect T-specific lymphocytes or cytokines/chemokines homing to within the primary site from the lymphatic system of the stomach can be attenuated by the eradication of HP through HPE. * HP examination: histologic examination, rapid urease test, ^13^C urea breath test, and serology examination.

**Table 1 cancers-14-01005-t001:** Published reports (more than 13 patients) on the efficacies of first-line antibiotics treatment in HP-negative gastric MALT lymphomas.

Author	Country	*N*	Stage	CR Rate *N* (%)	Median Time to CR: m (Range)	HP Test	HPE	t(11;18)	Ref.
Nakamura T, et al.	Japan	17	IE: 16 IIE1: 1	2 (11.8)	ND	H, RUT, S	A + C + P with/without M	CR: 1/2(+) Non-CR: 7/15(+)	[34]
Nakamura S, et al.	Japan	44	ND	6 (13.6)	ND	H, C, RUT, UBT, S	A + C + P C + M + P A + M + P	NA	[35]
Stathis et al.	Switzerland	14	IE: 9 IIE1: 5	5 (35.7)	ND	H, UBT, S	A + C + P, C + M + P, or A + M + P	NA	[36]
Asano et al.	Japan	17	IE: 15 IIE: 2	5 (29.4)	ND	H, RUT, UBT, S	A + C + P (16 pts) Or A + M + P (1 pt)	CR: 1/2(+), Non-CR: 6/9 (+)	[37]
Raderer et al.	Austria	13	IE: 8 IIE1: 5	5 (38.5)	3–36	H, UBT, S	C + M + P or C + A + P (7 or 14 D)	CR: 0/5(+), Non-CR: 3/8(+)	[30]
Gong et al.	Korea	28	IE: 24 IIE: 1 IV: 3	16 (57.1)	11.5 (10.0–22.3)	H, RUT, UBT, S	A + C + P 7D or 14 D	NA	[38]
Kuo et al.	Taiwan	25	IE: 22 IIE1: 3	8 (32.0)	6.1(1–24)	H, RUT, UBT, S, C	A + C + P 14 D	CR: 0/7(+) Non-CR: 6/13 (+)	[16]
Strati et al.	USA	26	IE: 26	6 (23.1)	2 (1–6)	H, S	A + C + P or M + C + P; 14 D	Non-CR: 0/3	[39]

Abbreviation: *N*, number; CR, complete remission; m, month; HP, Helicobacter pylori; HPE, HP eradication; t(11;18), t(11;18)(q21;q21); Ref, reference; ND, non-described; NA, non-analysis; pt, patient. HP examination test: H, histology; RUT, rapid urease test; UBT, urea breath test; S, serological test; C, culture; HPE regimen: A, amoxicillin; C, clarithromycin; M, metronidazole; P, proton-pump inhibitor, including lansoprazole, pantoprazole or esomeprazole for 7 to 14 days (D).

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
