# Peer review of "Current Status of the Spectrum and Therapeutics of Helicobacter pylori-Negative Mucosa-Associated Lymphoid Tissue Lymphoma"

_cancers, 2022, doi:10.3390/cancers14041005_

Round 1

Reviewer 1 Report

Authors summarize the recent studies for Helicobacter pylori-negative MALTG lymphoma. The article are well written, so only miner revision may be needed for acceptance.

Specific comment:

Authors indicated correlation between HP-negative gastric MALT lymphoma and HPE. In this connection, authors assume the mechanism as the eradication effect of unknown bacteria and/or indirect antineoplastic effect by clarithromycin. These assumptions are fully possibly. But the effect of HPE to NHPH may be some factor. So, authors may discuss in this point.

Old clinical studies take no thought of NHPH. It is likely that NHPH, especially urease positive bacteria, contaminates HP on the diagnosis of bacterial infection in the patients of MALT lymphoma. So, authors may discuss in this point if possible.

Minor comment:

In page 4, the fonts of the body in section 3 are incorrect.

Author Response

Authors’ response:

We appreciate the reviewer’s recommendations. According to your comments, we have provided new information in the subtitle “5. Possible molecular mechanisms of antibiotic-responsive HP-negative MALT lymphoma” (Page 7, Lines 336-340; Page 8, Lines 341-344), as follows: In addition to NHPH, other unknown bacterial sources can promote lymphomagenesis in HP-negative MALT lymphomas. Several hypotheses explain why certain patients with HP-negative MALT lymphoma respond to first-line HPE: (1) HPE regimens may eradicate NHPH, which is associated with the growth of HP-negative gastric MALT lymphoma in humans; (2) an HPE-like regimen may eradicate bacteria-associated antigens of unknown bacteria or unknown bacteria that may be associated with the lymphomagenesis of HP-negative MALT lymphoma; (3) clarithromycin, the principle component of the HPE regimen, may have direct or indirect antineoplastic effects, and thus cause CR of these tumors (Figure 1).

   In addition, we have provided the new information in the subtitle “3. Non-Helicobacter pylori Helicobacter (NHPH) may be associated with antibiotic-responsiveness of HP-negative gastric MALT lymphoma” (Page 5, Lines 214-234; Page 6, Lines 235-242) as follows:

Although most NHPHs are negative for the rapid urease test and 13C urea breath test, several studies have shown that certain NHPHs were positive for urease test and contaminated with HP in HP-positive patients [50,51,53]. Goji et al. reported that a patient presenting with nodular gastritis at the antrum tested positive for rapid urease test and a 13C urea breath test, but had negative results for stool antigen and serum anti-HP IgG antibody, and was further diagnosed with H. suis infection through PCR analysis of the 16S rRNA [50]. Goji et al. also reviewed the sensitivities of methods for detecting H. heilmannii-like organisms from 26 articles, and demonstrated that rapid urease test and immunohistochemical analysis were 40%, whereas urea breath test, blood antibody analysis, and stool antigen analysis were 14.8%, 23.1%, and 0%, respectively [50]. Yakoob et al. assessed the prevalence of NHPH and HP in 250 patients with dyspepsia through PCR analyses of Helicobacter genus-specific 16S rDNA, and reported that coinfection of H. heilmannii with HP was 6% in 17 patients and the coinfection of H. felis with HP was 4% in 10 patients [51]. In the assessment of the prevalence of H. heilmannii sensu lato (H. heilmannii s.l.), a group of NHPH species [52], including H. suis, H. felis, H. bizzozeronii, H. heilmannii sensu stricto (s.s.), and H. salomonis in samples of rapid urease test positive gastric biopsy using H. heilmannii s.l.-specific PCR followed by nucleotide sequencing, Liu et al. showed that NHPH infection was found in 178 (11.9%) of 1499 HP-positive patients [53]. Takigawa et al. also showed that 10% of patients with HP-positive MALT lymphoma had co-infection with NHPH, and NHPH-positive patients had more endoscopic nodular appearances in the gastric mucosa than NHPH-negative cases (3/16 [18.9%] vs. 0/29 [0%]) did [18]. These studies suggest that certain portions of NPHP may be involved in lymphomagenesis of urease test-positive patients, especially for patients with MALT lymphoma who presented with nodular appearances and negative serum antibody and stool antigen examination [18,50,54]. Although previous clinical studies exploring the efficacy of HPE in treating HP-positive gastric MALT lymphoma did not examine the prevalence of NHPH in these patients [6,7], further studies investigating the occurrence of NHPH in patients with MALT lymphoma presenting with nodular-like appearance irrespective of HP status are warranted.”.

We have corrected the fonts of the body in section 3 (page 4).

Reviewer 2 Report

The Authors have performed an exhaustive review concerning relationships between presentation and therapeutics of Helicobacter 2 pylori-negative mucosa-associated lymphoid tissue lymphoma. In particular, they  describe results obtained with first-line antibiotics, also taking into consideration the Helicobacter pylori (HP) eradication in patients affected by localized HP-unrelated MALT lymphoma and discussing the possible effective successful mechanisms. In addition,  for antibiotic-unresponsive HP-negative MALT  lym phoma, the application and efficacy of macrolides and immune modulatory drugs have been discussed as a therapeutic complement, other than conventional chemotherapy and immunochemotherapy.

The review is well structured with adequate subheadings and a good graphic scheme regarding Gastric and extra-gastric MALTomas; only a revision of English grammar and style of the paper should be requested before the publication in the Journal.

Author Response

Author’s response:

We appreciate your great efforts in reviewing our manuscript. We have carefully checked the grammar and spelling in the whole manuscript. We have improved our revised article (including English grammar, usage, and spelling) with copyediting service provided by a professional English language editing company (Editage, a division of Cactus Communications).

Reviewer 3 Report

The authors present an interesting, well written review about the background and the therapeutics of Helicobacter-negative MALT lymphoma. The efficacy of antibiotics in MALT lymphoma at different anatomical sites and in patients infected with different bacteria are described. Mechanisms underlying the antibiotice responses and immunological aspects are presented.

Under '6.' radiotherapy is mentioned as an option in "antibiotic-refractory HP-negative gastric MALT lymphoma" and "may induce acute and chronic adverse effects". In the following sentence the authors want to present an example but refer to "patients with OAML". The authors should present more adequately the important role of radiotherapy in HP-negative gastric MALT (and other anatomical sites) with continuously decreasing field size and doses of radiation in gastric MALT causing low and even lower adverse effects. They should refer to the largest published prospective analysis on radiotherapy in n=290 patients with gastric MALT (Reinartz, G., Pyra, R.P., Lenz, G. et al. Favorable radiation field decrease in gastric marginal zone lymphoma. Strahlenther Onkol 195, 544–557 (2019). https://doi.org/10.1007/s00066-019-01446-5) and modern biophysical analysis of side effects in current lower dose radiotherapy in gastric MALT

With regard to the above mentioned aspects, I recommend minor revision and resubmission of the manuscript.

Author Response

Reviewer 3:

The authors present an interesting, well written review about the background and the therapeutics of Helicobacter-negative MALT lymphoma. The efficacy of antibiotics in MALT lymphoma at different anatomical sites and in patients infected with different bacteria are described. Mechanisms underlying the antibiotice responses and immunological aspects are presented.

Under '6.' radiotherapy is mentioned as an option in "antibiotic-refractory HP-negative gastric MALT lymphoma" and "may induce acute and chronic adverse effects". In the following sentence the authors want to present an example but refer to "patients with OAML". The authors should present more adequately the important role of radiotherapy in HP-negative gastric MALT (and other anatomical sites) with continuously decreasing field size and doses of radiation in gastric MALT causing low and even lower adverse effects. They should refer to the largest published prospective analysis on radiotherapy in n=290 patients with gastric MALT (Reinartz, G., Pyra, R.P., Lenz, G. et al. Favorable radiation field decrease in gastric marginal zone lymphoma. Strahlenther Onkol 195, 544–557 (2019). https://doi.org/10.1007/s00066-019-01446-5) and modern biophysical analysis of side effects in current lower dose radiotherapy in gastric MALT

With regard to the above mentioned aspects, I recommend minor revision and resubmission of the manuscript.

The review is well structured with adequate subheadings and a good graphic scheme regarding Gastric and extra-gastric MALTomas; only a revision of English grammar and style of the paper should be requested before the publication in the Journal.

Author’s response:

We appreciate the reviewer’s suggestion. As your insightful comments, we have provided the new information on decreasing field size and doses of radiotherapy (RT) for localized HP-negative MALT lymphoma in the new subtitle section “6. Clinical efficacy of radiotherapy for HP-negative MALT lymphoma” (Page 9, Lines 430-438; Page 10, Lines 439-480) as follows:

For localized and antibiotic-refractory HP-negative gastric MALT lymphoma, radiotherapy (RT) can provide local control but may induce acute and chronic adverse effects [10,20,95,96]. For example, patients with OAML who received RT had a high CR rate and promising event-free survival (EFS) but experienced side effects, such as cataracts, dry eyes, and keratitis [95,97]. However, several studies have demonstrated that reduction in RT (field or dose) not only provides promising therapeutic efficacies, but also has lower adverse effects for localized MALT lymphoma [96,98,99,100,101]. In a long-term follow-up of 290 patients with stage IE-IIE HP-independent (antibiotic-refractory or HP-negative) gastric MALT lymphoma who received different RT fields from three consecutive prospective trials of the German Study Group on Gastrointestinal Lymphoma (DSGL), Reinartz et al. revealed that reduced RT field (stage I, involved field [IF] RT with 40 Gy; stage IIE locoregional extended field [EF] RT with 30 Gy followed by boost RT) provided similar EFS and overall survival (OS) when compared with EF RT (stage I, EF or reduced EF; Stage II, EF-mediastinum or EF; 30 Gy followed by 10 Gy boost) [98]. Furthermore, reduced RT field caused less acute hematological toxicities, such as anemia, leukocytopenia, and thrombocytopenia, and gastrointestinal toxicities, such as nausea and diarrhea, and lower frequencies of late toxicities than EF RT [98]. In a retrospective analysis of 178 patients with HP-independent gastric MALT lymphoma, who were stage I (86%), HP-negative status (80%), and most received involved-site RT (ISRT) with 30 Gy in 15 fractions, Yahalom et al. showed that among 160 patients who received regular panendoscopic examinations, 152 (95%) patients achieved CR, and most patients experienced rare acute toxicities, except for two patients who developed grade 3 toxicities [99]. In addition to the reduced RT field, Pinnix et al. revealed that ISRT with 24 Gy (n = 11) provided similar local control, freedom from treatment failure, and OS when compared with those receiving ISRT with 30 Gy (n = 21) for treating HP-independent gastric MALT lymphoma [100]. In a prospective trial comparing two different IF RTs for stage IE-II1E HP-independent gastric MALT lymphoma, Schmelz et al. showed that RT with 25.2 Gy (n = 10) produced a similar CR rate and gastrointestinal adverse effects when compared with RT with 36 Gy (n = 12) [101].

In the post-hoc analyses of a randomized, phase 3, non-inferiority trial comparing 24 Gy with 4 Gy for indolent lymphoma, including follicular lymphoma and MZL, authors reported that among 84 patients with MZL, patients (n = 41) receiving 24 Gy had a better 5-year local progression-free survival (PFS) than those receiving 4 Gy (n = 43) (100% vs. 88%, P = 0.015) [96]. Taken together, these findings indicate that reduced field RT with at least 24 Gy could provide promising locoregional control and long-term survival and favorable adverse effects. However, Pinnix et al. reported that 2 Gy in 2 fractions (total 4 Gy) resulted in a CR of 86% and only grade 1 eye syndrome in 5% of 22 patients with ocular adnexal lymphoma (OAL, including 14 with MALT lymphoma, 5 with follicular lymphoma, and 2 with mantle cell lymphoma) [102]. Cerrato et al. retrospectively analyzed 45 patients with MALT or MZL who were treated with 4 Gy in two 2-Gy fractions as curative or palliative intent, and revealed that low-dose RT (4 Gy) resulted in an ORR of 93 % (CR: 51%, PR: 42%) and a 2-year PFS and OS of 76% and 91%, respectively, without causing significant acute and late adverse effects [103]. Baron et al. also showed that among 36 patients with indolent OAL (20 patients with MALT lymphoma), low-dose RT (4 Gy) resulted in a similar ORR rate (100% [CR: 50%] vs. 87.5% [CR 58.3%]) but less acute toxicities (6/14 [42.9%] courses vs. 20/24 courses [83.3%], P = 0.014) when compared with moderate-dose RT (21–36 Gy) [104]. Prospective studies to validate the local control rate and adverse effects of low-dose RT (4 Gy) in larger patients with MALT lymphoma are warranted.

  In addition, we have carefully checked the grammar and spelling in the whole manuscript. We have improved our revised article (including English grammar, usage, and spelling) with copyediting service provided by a professional English language editing company (Editage, a division of Cactus Communications).
